# Association between Productive Roles and Frailty Factors among Community-Dwelling Older Adults: A Cross-Sectional Analysis

**DOI:** 10.3390/ijerph191710838

**Published:** 2022-08-31

**Authors:** Kazuki Yokoyama, Hikaru Ihira, Yuriko Matsuzaki-Kihara, Atsushi Mizumoto, Ryo Miyajima, Takeshi Sasaki, Naoki Kozuka, Nozomu Ikeda

**Affiliations:** 1Department of Occupational Therapy, School of Health Sciences, Sapporo Medical University, Sapporo 060-8556, Japan; 2Department of Physical Therapy, School of Health Sciences, Sapporo Medical University, Sapporo 060-8556, Japan; 3Department of Rehabilitation, Japan Health Care University, Sapporo 062-0053, Japan; 4Department of Physical Therapy, Faculty of Human Science, Hokkaido Bunkyo University, Eniwa 061-1449, Japan; 5Ebetsu City Hospital, Ebetsu 067-8585, Japan

**Keywords:** older adults, productive roles, work, employment, frailty

## Abstract

The employment rate of older people in Japan is expected to increase in the future owing to the increase in the retirement age. Preventing frailty is imperative to maintaining productive roles of older adults. Therefore, this study aimed to examine the association between productive roles and frailty factors among community-dwelling older adults. A total of 135 older adults, enrolled in 2017, participated in the study. Productive roles and domains related to frailty were measured. We measured usual gait speed and grip strength for the physical domain; Mini-Mental State Examination (MMSE) and Geriatric Depression Scale (GDS-15) scores for the cognitive and mental domains; and social role and group activity for the social domain. Multivariate-adjusted logistic regression models revealed that having productive roles was associated with faster usual gait speed (odds ratios [OR] = 1.05; 95% confidence interval [CI], 1.01–1.08; *p* = 0.005) and lower GDS-15 score (OR = 0.79; 95% CI, 0.64–0.97; *p* = 0.023). These results suggest that health promotion to maintain gait speed and prevent depressive symptoms may contribute to maintaining productivity in community-dwelling older adults.

## 1. Introduction

In the next half century, the world will experience rapid population aging in both developed and developing regions. The percentage of the population aged ≥65 years is expected to increase from 17.6% to 28.2% in developed regions and from 6.3% to 16.4% in developing regions, between 2015 and 2060 [1]. The World Health Organization (WHO) considers healthy aging as a basic framework for designing public health strategies for creating environments and opportunities that enable people to be and do what they value throughout their lives [2]. Healthy aging is defined as the process of developing and maintaining functional abilities that enable well-being in older age [2]. It depends on the interrelationship with the environment and external world that forms the context of an individual’s life, such as social networks, support, participation, and activity [3].

One way in which the older adults interact with society is through work. In Japan, the employment rate of the population aged 65 years was 25.1% in 2021 and is increasing annually [4]. This rate is higher compared to the rates of other major countries such as the United States (18.0%), Canada (12.8%), the United Kingdom (10.5%), and Germany (7.4%) [5]. One of the reasons for the increase in the employment rate among the older population is the increase in Japan’s aging rate, which is the highest worldwide at 28.4% [6]. The Japanese government has taken measures to promote employment of older adults to secure the dwindling workforce as the working-age population declines. Many Japanese companies employ mandatory retirement, a system in which employment contracts are terminated upon reaching a certain age, and few people work beyond the retirement age. However, in April 2021, the Revision of the Act on Stabilization of Employment of Elderly Persons [7] resulted in an increase of the retirement age from 60 years to 65 years, allowing workers aged 65–70 years to continue working if they wish. Based on the above, a workforce comprising the older adults is expected to increase further, resulting in an increase in the need for healthy aging.

For older adults, working is beneficial not only from the perspective of increasing tax revenues and reducing pensions, but also for achieving a meaningful life. Older adult workers make proactive efforts to maintain or adapt to high levels of productivity [8]. In Japan, nearly 40% of working older people aged ≥60 years reported that they wished to work for as long as they could work, and nearly 90% of the same population wished to continue working until the age of ≥70 years [6]. Willingness to work has positive effects on individual health. In a study comprising Japanese men, individuals employed in old age lived for 1.91 years longer and delayed cognitive decline by 2.22 years compared with those not employed [9]. Less productive roles, such as paid work and housework, were related to more depressive symptoms, although volunteer work attenuated the negative effect of losing their roles [10]. Thus, retirement has a negative causal impact on cognitive functioning [11]. Conversely, continued employment is assumed to maintain job stress and occupational hazards and decrease the time spent engaging in leisure activities. A systematic review indicated evidence that retirement is effective in improving mental health; however, perceived general health and physical health have shown inconsistent results [12].

Among the various findings on continuing employment for older adults, in addition to paid employment, it is likely that continuing social roles and social participation through engagement in productive roles is useful for healthy aging [8,9,10]. Therefore, engagement in productive roles as a trigger for social participation in old age may assist in the prevention of frailty. Several previous studies indicated that social frailty was associated with depressive symptoms [13] in addition to physical and cognitive functions [14]. However, few studies have examined the engagement of older adults in productive roles, including paid (full-time or part-time), unpaid, and volunteer work. Since frailty confers high risk for adverse health outcomes [15,16], it is assumed that older adults should avoid frailty to continue their productive roles. Frailty is divided into three domains: physical, cognitive and psychological, and social frailty [15]. However, factors of frailty related to continuation of their productive roles have not been identified. Identifying associations between productive roles and the three frailty domains is likely to be useful for interventions related to frailty control and care prevention to maintain employment. This cross-sectional study aimed to examine the association between productive roles and frailty factors among community-dwelling older adults.

## 2. Materials and Methods

### 2.1. Participants

This cross-sectional study used data collected in 2017 from the Widely Hokkaido Individual Training for Elderly (WHITE) study. The eligibility criterion for participation in the WHITE study was as follows: individuals aged ≥65 years who attended the survey meeting in Sapporo, Hokkaido, Japan. A total of 182 individuals were included in this survey. Information on demographics, productive roles, depressive symptoms, and social domains was obtained from face-to-face interviews and questionnaires. Subsequently, measures of physical and cognitive functions were performed. The exclusion criteria were as follows: (1) history of stroke, dementia, mild cognitive impairment, and other neurological disorders, (2) need for support or care as certified by the Japanese public long-term care insurance system, and (3) insufficient data regarding demographics and assessment of the three domains. At the end of the exclusion process, 135 participants (56 men and 79 women) were included in this study.

### 2.2. Assessment of Exposure and Outcomes

#### 2.2.1. Demographics

Demographic data, including age, sex, years of education, living alone status, and subjective economic status, were collected through face-to-face interviews and questionnaires. Living alone status was identified using the yes/no method, and subjective economic status was identified using the following options: enough and not worried; not enough, but not worried; and not enough and worried.

#### 2.2.2. Productive Roles

The participants were asked whether they were engaged in productive roles using the yes/no method. Productive roles were defined as engagement in any activity that produced goods and/or services, including paid (full-time or part-time), unpaid, and volunteer work. This operational definition has been used in a previous Japanese study [10] and is based on the review of several empirical studies [17,18,19].

#### 2.2.3. Physical Frailty

Physical frailty was determined using usual gait speed and grip strength, as measured by the Cardiovascular Health Study index [16]. Usual gait speed was measured as the time required to walk at a normal pace along a 2.4 m walking path with a 2-m preliminary section each, at the beginning and end (Walk Way MW-1000; Anima, Tokyo, Japan) [20]. Participants were instructed to walk at the speed at which they normally walked. Grip strength was measured using a Smedley dynamometer (Matsumiya Medical Industry, Tokyo, Japan). Two trials were performed with the dominant hand and the larger value was recorded [21].

#### 2.2.4. Cognitive and Psychological Frailty

Cognitive decline comprised cognitive and psychological frailty and depressive symptoms [22]. Cognitive function was measured using the Japanese version of the Mini-Mental State Examination (MMSE) [23], which is widely used internationally as a reliable and valid test to assess cognitive function and screen for cognitive impairment [23,24]. The MMSE scores range from 0 to 30, with lower scores indicating weaker cognitive function. Geriatric depression was assessed using the Geriatric Depression Scale (GDS-15) [25,26], which has been validated for screening depressive symptoms in older people [25,26,27]. The GDS-15 is a self-reported questionnaire consisting of 15 binary items with yes/no responses. The total score ranges from 0 to 15, with higher scores indicating more depressive symptoms.

#### 2.2.5. Social Frailty

Social frailty is screened by various factors, such as going out, visiting friends, and feeling helpful to friends or family [28]. In this study, social roles and social participation were incorporated because they were associated with productive roles. Social role was assessed using the subscale of the Tokyo Metropolitan Institute of Gerontology Index of Competence [29], which is a widely used reliable and valid scale to measure competence. For the social role subscale, four items were self-rated using yes/no responses. The total score ranges from 0 to 4, with higher scores indicating a social role. The social participation of the participants was assessed whether they usually participate in groups, such as senior citizen clubs, circles for hobbies, sports, learning, civic groups, and local governments based on a previous study [30]. The options for this part were as follows: “not participating”, “once a year or less”, “several times a year”, and “once a month or more”. Frequency of participation in group activities has been shown to be associated with maintaining functional competence for 4 years [30]. In this study, “not participating” and “once a year or less” indicated low participation, and “several times a year” and “once a month or more” indicated high participation.

### 2.3. Data Analysis

Participants who had productive roles were categorized into the “Productive Roles (PR)” group and those who did not have productive roles into the “non-PR” group. Participant characteristics were calculated as means and standard deviations (SDs) for continuous variables, and numbers and percentages (%) for categorical variables in the two groups. Student’s t-tests for continuous variables and the χ^2^ test for categorical variables were tested to clarify the differences between groups. Logistic regression analysis was performed with productive roles (non-PR = 0, PR = 1) as the dependent variable and each variable of the three domains as the independent variable. The odds ratio (OR) and 95% confidence intervals (95% CIs) for productive roles according to frailty were calculated in crude form. Subsequently, two models adjusted for age and sex, with education, living alone status, and subjective economic status as covariates were examined. Next, stratified analyses were performed by age category (young-old adults aged 65–74 years; old-old adults aged 75 years and older). This analysis also used the crude model and two adjusted models to calculate OR and 95% CI. The SPSS version 27.0 (IBM Inc., Armonk, NY, USA) was used for analysis, and the level of significance was set at 0.05.

### 2.4. Ethical Considerations

This study was approved by the Sapporo Medical University Ethical Review Board (approval number 28-2-7). Written informed consent was obtained after the procedures had been fully explained to each participant. Their anonymity was consistently preserved.

## 3. Results

The mean age of the 135 (93 [68.9%], non-PR group; 42 [31.1%], PR group) participants was 74.3 (SD = 5.5) years. The participants’ characteristics are summarized in Table 1. There were no significant group differences in age, sex, years of education, and living alone status. However, subjective economic status had significant differences between the non-PR and PR groups (χ^2^ = 7.360, *p* = 0.025). In the variables related to frailty, the PR group had significantly faster usual gait speed (t = −2.175, *p* = 0.007) and lower GDS-15 score (t = 2.360, *p* = 0.020) compared to the non-PR group. However, there were no significant differences in the other variables between the two groups.

The results of logistic regression analyses are presented in Table 2. Compared with participants in the non-PR group, the PR group was significantly associated with usual gait speed (OR = 1.05; 95% CI, 1.01–1.08; *p* = 0.005) and GDS-15 score (OR = 0.79; 95% CI, 0.64–0.97; *p* = 0.023) after adjusting for age, sex, years of education, living alone status, and subjective economic status. There was no significant difference in grip strength, MMSE, social role, and group activity between the two groups (*p* > 0.05).

The results of the stratified analysis for young-old adults (*n* = 80, 58.8% women) and old-old adults (*n* = 55, 58.2% women) are shown in Table 3. The mean ages were 70.5 ± 2.4 and 79.8 ± 3.8 years in young-old and old-old adults, respectively. Among young-old adults, compared with participants in the Non-PR group, the PR group was significantly associated with slower usual gait speed (OR = 1.06, 95% CI, 1.01–1.12; *p* = 0.021) and GDS-15 score (OR = 0.70; 95% CI, 0.51–0.95; *p* = 0.024) after adjusting for adjusting for age, sex, years of education, living alone status, and subjective economic status. Among old-old adults, no significant association was found for these variables, whereas the PR group was significantly associated with group activity (OR = 13.61; 95% CI, 1.10–169.19; *p* = 0.042) after adjusting for these covariates.

## 4. Discussion

This study examined the association between productive roles and frailty factors among community-dwelling older adults. We found that among the various frailty factors, in community-dwelling older adults, faster gait speed and lower depression score were associated with having productive roles.

Regarding the association between physical frailty and productive roles, the participants who had productive roles had faster usual gait speed but had no association for grip strength. Similar trends were observed in young-old adults. In the revised Japanese version of the Cardiovascular Health Study criteria, the criterion for healthy gait speed is ≤1.0 m/s [31]. Most of the participants in this study exceeded this criterion for gait speed, but there were significant differences in the employment role. Thus, the results of the present study suggest that differences in gait speed above the criterion level can predict productive roles. A previous study showed that faster gait speed was associated with higher physical activity; however, grip strength was not associated with physical activity [32]. Physical activity and leg strength were independent predictors of the decline in mobility performance [33]. Therefore, the mobility of the lower limbs, which reflects gait speed, was identified as an important factor in productive roles. Maintaining muscle strength, especially in the lower extremities, may help maintain productive roles because it encourages physical activity. In contrast, it can be hypothesized that older adults have higher physical activity and faster gait speed because of having productive roles. Since this is an important perspective in the prevention of frailty, it is necessary to examine the causal relationship between gait speed and productive roles in the future.

From the perspective of psychological and cognitive frailty, geriatric depression was found to be a predictor of productive role among the young-old adults. A previous study found that geriatric depression was an independent variable for social frailty [13]. Social frailty was defined as living alone, talking with someone every day, feeling helpful to friends or family, going out less frequently compared with the previous year, and visiting friends sometimes [28]. It is hypothesized that people with lower depressive symptoms are less likely to experience social frailty because they remain more motivated, which may lead to maintenance of their productive roles, than those with higher depressive symptoms. In contrast, less productive roles and social frailty were identified as independent variables for the incidence of depressive symptoms [10,34]. Retirement may cause withdrawal from social participation and connection to society, and this negatively affects higher-level functional capacity [35]. In Japan, healthy life expectancy increased to 72.14 and 74.79 years for men and women in 2016, respectively [36], and social participation has expanded especially among the older adults in the first half of their lives. In these situations, it is possible that the early loss of productive roles associated with retirement and withdrawal from society may have augmented depressive symptoms and led to lesser social participation among the young-old adults. Thus, productive roles and depressive symptoms are likely to be interrelated, and depressive symptoms need to be controlled to continue social participation and, eventually, productive roles. In addition, cognitive function did not predict productive role. Cognitive frailty is defined as the presence of both cognitive decline and physical frailty and shows an increased risk of limitations in the instrumental activities of daily living [22]. However, similar to the present study, cognitive function measured using the MMSE has little correlation with social frailty in community-dwelling older adults [37]. This result may be influenced by the fact that the participants in this study were able to participate in the survey session invitations on their own, and there were few participants who had both physical frailty and cognitive decline.

Furthermore, regarding social frailty and productive roles, an association was found between participation in group activities and productive roles among the old-old adults. This may indicate that those with connection to the community through group activities are more likely to maintain employment-related roles. In Japan, a basic checklist for care prevention tailored to the characteristics of old-old adults has been developed, which includes items on social participation and support [38]. However, this checklist includes the following questions: “Do you go out at least once a week?” and “Do you usually socialize with family and friends?” However, it did not include questions related to participation in group activities. According to the socioemotional selectivity theory, it is possible for older adults to be emotionally-satisfied by maintaining interactions centered on close relationships as they experience various stages of aging [39]. For old-old adults, it was highly likely that participation in group activities, even if only several times a year, was associated with maintaining or acquiring productive roles. It could lead to the positive emotional impact.

This study has some limitations. This cross-sectional study was conducted in a specific region of Japan. Further longitudinal studies are required to identify the factors that influence the maintenance of productive roles. In addition, productive roles are a broadly defined concept that includes numerous forms of activity, such as paid (full-time or part-time), unpaid, and volunteer work. Based on these differences, the factors associated with frailty may differ. Simultaneously, it is probable that the meaning of productive roles will vary among participants. For example, some people work as an obligation to earn a wage, whereas others work because they wish to add meaning to their life. Further studies should therefore be conducted to categorize the meaning of work based on these factors.

## 5. Conclusions

This study investigated frailty factors associated with engagement in productive roles. We identified faster gait speed and lower depressive symptoms as frailty factors that were associated with productive roles. The stratified analyses showed a similar trend in young-old adults. Moreover, for older adults, participation in group activities several times a year and more was significantly associated with productive roles. However, this result needs to be clarified in future longitudinal studies. We propose that preventive and rehabilitative interventions should be developed to help older adults continue meaningful social participation through productive roles.

## Figures and Tables

**Table 1 ijerph-19-10838-t001:** Comparisons of characteristics.

	Variable	Non-PR (*n* = 93)	PR (*n* = 42)	*p* Value
Demographics	Age, years	74.2 ± 5.3	74.4 ± 5.9	0.827
	Sex, women	54 (58.1%)	25 (59.5%)	0.873
	Education, years	13.2 ± 2.1	13.3 ± 2.6	0.784
	Living alone status	20 (21.5%)	9 (22.0%)	0.992
	Subjective economic status			
	Enough and not worried	7 (7.5%)	7 (16.7%)	0.025 *
	Not enough, but not worried	79 (84.9%)	27 (46.3%)	
	Not enough and worried	7 (7.5%)	8 (19.0%)	
Physical domains	Usual gait speed, m/min	80.4 ± 12.2	86.7 ±13.2	0.007 **
	Grip strength, kg	26.3 ± 9.8	26.9 ± 9.1	0.738
Cognitive and psychological domains	MMSE (0–30)	27.9 ± 2.1	28.0 ± 2.2	0.840
	GDS-15 (0–15)	2.8 ± 2.2	1.9 ± 2.0	0.020 *
Social domains	Social role (0–4)	0.8 ± 0.9	0.6 ± 0.7	0.407
	Group activity			
	No participation/Once a year or less	21 (22.6%)	4 (9.5%)	0.080
	Several times a year/Once a month or more	72 (77.4%)	38 (90.5%)	

Mean ± standard deviation or *n* (%), the Student’s *t*-test (age, education, usual gait speed, grip strength, MMSE score, GDS-15 score, and social role), the χ^2^ test (sex, living alone status, subjective economic status, and group activity), * *p* < 0.05, ** *p* < 0.01.

**Table 2 ijerph-19-10838-t002:** Odds ratios and 95% confidence intervals for productive roles according to frailty.

**Physical Domains**	**Usual Gait Speed**		**Grip Strength**	
	**Non–PR**	**PR**	**Non–PR**	**PR**
OR 1 (95% CI)	1	1.04 (1.01–1.08) *	1	1.01 (0.97–1.05)
OR 2 (95% CI)	1	1.05 (1.01–1.08) *	1	1.02 (0.97–1.08)
OR 3 (95% CI)	1	1.05 (1.01–1.08) *	1	1.02 (0.97–1.08)
**Cognitive and** **Psychological Domains**	**MMSE**		**GDS–15**	
	**Non–PR**	**PR**	**Non–PR**	**PR**
OR 1 (95% CI)	1	1.02 (0.86–1.21)	1	0.79 (0.64–0.97) *
OR 2 (95% CI)	1	1.03 (0.86–1.23)	1	0.79 (0.64–0.97) *
OR 3 (95% CI)	1	1.03 (0.85–1.23)	1	0.79 (0.64–0.97) *
**Social Domains**	**Social Role**		**Group Activity**	
	**Non–PR**	**PR**	**Non–PR**	**PR**
OR 1 (95% CI)	1	0.83 (0.53–1.29)	1	2.77 (0.89–8.66)
OR 2 (95% CI)	1	0.83 (0.52–1.31)	1	2.76 (0.88–8.63)
OR 3 (95% CI)	1	0.82 (0.51–1.30)	1	2.87 (0.91–9.10)

OR 1 = crude, OR 2 = adjusted for age and sex, OR 3 = adjusted for age, sex, years of education, living alone status, and subjective economic status, * *p* < 0.05.

**Table 3 ijerph-19-10838-t003:** Odds ratios and 95% confidence intervals for productive roles according to frailty by age category.

**Young-old adults (*n* = 80)**	**Physical domains**	**Usual gait speed**		**Grip strength**	
		**Non-PR**	**PR**	**Non-PR**	**PR**
	OR 1 (95% CI)	1	1.06 (1.01–1.12) *	1	1.00 (0.95–1.04)
	OR 2 (95% CI)	1	1.06 (1.01–1.11) *	1	1.06 (0.98–1.16)
	OR 3 (95% CI)	1	1.06 (1.01–1.12) *	1	1.08 (0.98–1.18)
	**Cognitive and** **psychological domains**	**MMSE**		**GDS–15**	
		**Non–PR**	**PR**	**Non–PR**	**PR**
	OR 1 (95% CI)	1	1.13 (0.85–1.49)	1	0.71 (0.53–0.95) *
	OR 2 (95% CI)	1	1.10 (0.82–1.46)	1	0.69 (0.51–0.94) *
	OR 3 (95% CI)	1	1.10 (0.78–1.55)	1	0.70 (0.51–0.95) *
	**Social domains**	**Social role**		**Group activity**	
		**Non–PR**	**PR**	**Non–PR**	**PR**
	OR 1 (95% CI)	1	0.70 (0.38–1.28)	1	1.71 (0.43–6.79)
	OR 2 (95% CI)	1	0.79 (0.43–1.47)	1	1.86 (0.39–1.86)
	OR 3 (95% CI)	1	0.79 (0.41–1.52)	1	1.96 (0.45–8.58)
**Old–old adults (*n* = 55)**	**Physical domains**	**Usual gait speed**		**Grip strength**	
		**Non–PR**	**PR**	**Non–PR**	**PR**
	OR 1 (95% CI)	1	1.03 (0.99–1.07)	1	1.04 (0.97–1.12)
	OR 2 (95% CI)	1	1.04 (0.99–1.09)	1	1.01 (0.92–1.11)
	OR 3 (95% CI)	1	1.03 (0.98–1.09)	1	1.03 (0.93–1.15)
	**Cognitive and** **psychological domains**	**MMSE**		**GDS–15**	
		**Non–PR**	**PR**	**Non–PR**	**PR**
	OR 1 (95% CI)	1	0.96 (0.75–1.22)	1	0.88 (0.68–1.16)
	OR 2 (95% CI)	1	0.99 (0.77–1.28)	1	0.83 (0.63–1.09)
	OR 3 (95% CI)	1	1.00 (0.76–1.33)	1	0.84 (0.62–1.15)
	**Social domains**	**Social role**		**Group activity**	
		**Non–PR**	**PR**	**Non–PR**	PR
	OR 1 (95% CI)	1	1.07 (0.54–2.10)	1	6.30 (0.74–53.69)
	OR 2 (95% CI)	1	0.94 (0.46–1.90)	1	11.08 (1.13–108.91) *
	OR 3 (95% CI)	1	1.03 (0.48–2.24)	1	13.61 (1.10–169.19) *

OR 1 = crude, OR 2 = adjusted for age and sex, OR 3 = adjusted for age, sex, years of education, living alone status, and subjective economic status, * *p* < 0.05

## Data Availability

The datasets generated and/or analyzed during the current study are available from the corresponding author upon reasonable request.

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
