# Peer review of "Association between Productive Roles and Frailty Factors among Community-Dwelling Older Adults: A Cross-Sectional Analysis"

_ijerph, 2022, doi:10.3390/ijerph191710838_

Round 1

Reviewer 1 Report

Overall, this paper was not well explained and it was difficult to understand the quality of the content.

First of all, it seems obvious that the frail elderly tended to not do working or volunteer activities (because frailty is concept to predict functioning and disability in activities of daily living originally). The authors also misunderstand the concept of productive role. The authors mentioned their logic as if all older adults must engage in work and volunteer activities, but it is a matter of personal preference whether or not to engage in these activities at retirement age, even if they are robustly healthy. For example, it may be more important to elucidate the characteristics of those who want to work but cannot. It is difficult to understand the importance of the authors' hypothesis. The concepts of both frailty and productive roles are vague.

There was not a sufficient description for what indicator was used to evaluate productive roles and whether the indicator is valid.

The methods and results of statistical analyses are not sufficiently described.

Reviewer 2 Report

At a time when the number of elderly people is increasing dramatically all over the world, it is interesting and relevant to know the association between productive roles and frailty factors among community-dwelling older adults.

Your topic is very interesting, although you should improve your article. I hope you can do this.

You should deepen your description of the method, including a description of all the instruments used and clarify whether all were previously validated for your population, as well as deepen the discussion and conclusions. Ethical procedures must also be clearly referred to and explained.

You should give tables a title. Correct the error.

Round 2

Reviewer 1 Report

Unfortunately, the authors did not adequately respond to the previous comments.

Author Response

Thank you very much for your comments. We have added that the measurement variables are valid in our population, citing previous studies.  We would appreciate your review of the revised manuscript.

Reviewer 2 Report

Thank you for your review and your efforts to improve the work. I believe you really did.

line 142: "This operational definition has been used in previous Japanese study [10] and is based on the review of several empirical studies" (which are?)

It remains unclear whether all instruments were previously validated for your population This needs to be clear. Please clarify.

Author Response

We thank the reviewer for carefully reading our manuscript and providing useful comments. We appreciate your review of the attached files.
